# Deterrent Effects of Clary Sage Oil and Two Major Constituents against *Drosophila suzukii* (Diptera: Drosophilidae)

**DOI:** 10.3390/insects15100733

**Published:** 2024-09-24

**Authors:** Yu Wang, Fengyi Wen, Xiangyi Zhou, Guoxing Chen, Chunxia Tian, Jiali Qian, Huiming Wu, Mengli Chen

**Affiliations:** 1Zhejiang Key Laboratory of Biology and Ecological Regulation of Crop Pathogens and Insects, College of Advanced Agricultural Sciences, Zhejiang A&F University, Hangzhou 311300, China; wy030124@163.com (Y.W.); 15500926573@163.com (F.W.); 17767273866@163.com (X.Z.); guoxingchen@stu.zafu.edu.cn (G.C.); wuhm@zafu.edu.cn (H.W.); 2Zhejiang Provincial Center for Disease Control and Prevention, Hangzhou 310051, China; cxtian@cdc.zj.cn; 3Institute of Pesticide and Environmental Toxicology, Zhejiang University, Hangzhou 310030, China; jialiqian@zju.edu.cn

**Keywords:** spotted-wing drosophila, clary sage oil, linalyl acetate, linalool, repellency, EAG

## Abstract

**Simple Summary:**

*Drosophila suzukii* (Diptera: Drosophilidae) is an invasive pest that lays eggs in ripe fruits. The short generation time, high fertility, and wide host range allows this species to cause important yield losses. Essential oils, extracted from a number of plant species, have been studied for their attractiveness to and deterrence of *D. suzukii*. In this study, we found clary sage oil exhibited dose-dependent repellency against both *D. suzukii* adults and larvae. Also, we identified that linalyl acetate and linalool were two major constituents of clary sage oil by GC-MS. Furthermore, we detected that both linalyl acetate and linalool repelled *D. suzukii* adults and larvae. In addition, we examined the electroantennography (EAG) responses of *D. suzukii* to clary sage oil, linalyl acetate, and linalool. These results suggest clary sage oil and its constituents linalyl acetate and linalool could be potential repellents for the management of *D. suzukii*.

**Abstract:**

*Drosophila suzukii* (Diptera: Drosophilidae), spotted-wing drosophila, poses a significant threat to soft-skinned fruit crops in the Americas, Europe, Africa, and Oceania, as well as in Asia. The application of chemical insecticides is the primary control strategy for *D. suzukii*; however, resistance has developed with the indiscriminate use of chemical insecticides. Essential oils, considered potential alternatives to pesticidal strategies, exhibit potent toxic and sublethal behavioral effects against numerous pests, including *D. suzukii*. Clary sage oil repels a variety of agricultural and household pests; however, whether it has a repellent effect against *D. suzukii* remains unknown. Here, we found that clary sage oil exhibited dose-dependent repellency against *D. suzukii* adults in a T-maze assay, a two-choice assay and a two-choice attraction assay. Also, clary sage oil showed a significant repellent effect against *D. suzukii* larvae. Next, we explored the chemical constituents of clary sage oil by GC-MS and identified two major constituents, linalyl acetate (40.03%) and linalool (23.02%). Furthermore, the behavioral assays of linalyl acetate and linalool showed that both compounds conferred comparable repellency against *D. suzukii* adults and larvae. Finally, we found clary sage oil, linalyl acetate, and linalool elicited EAG responses in *D. suzukii*, especially clary sage oil, suggesting the repellency was mediated by the olfactory system. These findings indicate that *D. suzukii* shows olfactory-based behavioral avoidance of clary sage oil, linalyl acetate, and linalool. Clary sage oil and its major constituents may be possible alternatives in the management of *D. suzukii*.

## 1. Introduction

The spotted-wing drosophila (*Drosophila suzukii* Matsumura), an invasive fruit fly species, has garnered considerable attention due to its detrimental impact on a variety of fruit crops, such as berries, cherries, grapes, and many other soft-skinned fruits [1]. *D. suzukii*, originating in East Asia, has spread to other parts of the world, including Europe, the Americas, Africa, and Oceania, where it has become a significant threat to the fruit industry [2]. *D. suzukii* punctures fruits with its serrated ovipositor to lay eggs, and the larvae that emerge from these eggs feed on the fruit, causing extensive damage that renders it unmarketable [3].

Currently, *D. suzukii* management is primarily reliant on chemical insecticides, such as organophosphates, pyrethroids, spinosyns, and neonicotinoids. However, the frequent use of chemical insecticides has led to the rapid development of resistance and caused problems due to chemical residues; thus, it is urgent to develop more sustainable and environmentally friendly alternative solutions [4,5]. The application of essential oils, extracted from various aromatic plants, has emerged as a promising strategy for controlling *D. suzukii*, according to multiple investigations in the literature [6,7]. Previous studies have reported that multiple essential oils have shown direct toxicity to the larvae and adults of *D. suzukii* [8]. Moreover, lots of essential oils have demonstrated the ability to disrupt the olfactory responses of *D. suzukii*, thereby repelling them from host fruits [9].

Essential oils are a complex mixture consisting of a diverse array of chemical components, including monoterpenes, sesquiterpenes, phenylpropanoids, and other volatile constituents [10]. There is little doubt that the bioactive components of essential oils, especially mono- and sesquiterpenoids, contribute to the toxic effects of essential oils observed in insects [11,12]. For instance, as the main constituents of essential oils from *Citrus aurantiifolia* (Chrismann) Swingle and *Citrus reticulata* Blanco, both (R)- and (S)-limonene caused significant contact toxicity against *Sitophilus zeamais* Motschulsky (Coleoptera: Curculionidae) [13]. Moreover, these active compounds also display potentially important sublethal behavioral effects in pests, including feeding and oviposition deterrence and repellence [13]; for example, citronella oil, a widely used natural mosquito repellent, also deterred other pests, such as *Tribolium castaneum* (Herbst) (Coleoptera: Tenebrionidae) [14] and *Cochliomyia hominivorax* Coquerel (Diptera: Calliphoridae) [15]. 

Clary sage oil, derived from *Salvia sclarea* Linnaeus (Lamiaceae), is renowned for its diverse applications in aromatherapy, cosmetics, and medicine [16,17]. Beyond its aromatic and skincare uses, clary sage oil has demonstrated potential as a natural insect repellent and insecticide. Previous studies have shown that clary sage oil exhibits insecticidal activity against *S. zeamais* [18], *Aedes aegypti* Linnaeus (Diptera: Culicidae) [19], *Thrips flavus* Schrank (Thysanoptera: Thripidae) [20], and many other pests. In addition, clary sage oil has been demonstrated to deter a variety of agricultural and household pests, such as *T. castaneum* [21], *Tetranychus urticae* Koch (Acari: Teranychidae) [22], *Lasioderma serricorne* Fabricius (Coleoptera: Anobiidae) [23], and *Musca domestica* Linnaeus (Diptera: Muscidae) [24]. 

Although clary sage oil exhibits excellent repellent effects on various kinds of pests, whether clary sage oil repels *D. suzukii* is still unknown. Here, we conducted several behavioral assays to examine the repellent effect of clary sage oil on *D. suzukii* adults and larvae. Furthermore, to explore whether the major constituents confer the repellency of clary sage oil, we analyzed the constituents of clary sage oil and examined the repellent effect of the major constituents, linalyl acetate and linalool. Lastly, we explored the EAG responses of *D. suzukii* adults to clary sage oil and its two major constituents. This study illustrates that clary sage oil and its major constituents might have great potential for the management of *D. suzukii*.

## 2. Materials and Methods

### 2.1. Insect Rearing

*D. Suzukii*, kindly provided by Shaoying Wu (Hainan University), were collected from the field in Haikou, Hainan Province, China (20° N, 110° E) in 2018, and were reared in laboratory conditions for more than five years. *D. suzukii* were reared on standard cornmeal–yeast agar medium (Genesee Scientific, San Diego, CA, USA) with poplar wood wool (Hualeji, Hangzhou, China) and filter paper (Jiaojie, Dongguan, China). *D. suzukii* were maintained in a growth chamber at 25 ± 1 °C under 60% RH with a light/dark photoperiod of 12 h each. 

### 2.2. Essential Oils and Chemicals 

Clary sage oil was purchased from Dr Wong Co., Ltd. (Dr Wong, Guangzhou, China). Two constituents of clary sage essential oil, linalyl acetate and linalool, were purchased from Tokyo Chemical Industry (TCI, Tokyo, Japan). The purities of these two compounds were above 95%. 

### 2.3. T-Maze Assay

The T-maze apparatus was set up with three arms, two of which were perpendicular to each other (forming the “T” shape), and a central starting area, as shown in Figure 1A. The two arms were formed by microcentrifuge tubes and Tygon tubing, connected by pipette tips, and the central starting area was formed by pipette tips. In both microcentrifuge tube lids, a small hole was made to let air flow through. The filter paper was applied with 50 μL of odorants or solvent control (paraffin oil) and lined the wall of the microcentrifuge tube. Three- to five-day-old *D. suzukii* flies (10 males and 10 females) were gently introduced into the Tygon tubing through the small hole made in the middle of the tubing. The assay was run at 25 °C with a light/dark photoperiod of 12 h each. The number of flies entering each arm was counted after 24 h. The preference index (PI) was calculated as (O−C)/(O+C), where O is the number of flies in the test odorant trap, and C is the number of flies in the solvent control trap.

### 2.4. Two-Choice Assay

The two-choice assay was identical to those described previously [25]. Two traps were made with microcentrifuge tubes and pipette tips, and the filter paper with 10 μL of the odorants or solvent control applied was lined in the wall of pipette tips. Two traps were secured in a 100 mL glass beaker using double-sided tape (Figure 1C). Three- to five-day-old *D. suzukii* were used in this assay, and forty to fifty flies (half males and half females) were gently introduced into the glass beaker and covered with gauze. The assay was run at 25 °C with a light/dark photoperiod of 12 h each. The number of flies entering each trap was counted after 24 h. The preference index (PI) was calculated as in the T-maze assay.

### 2.5. Two-Choice Attraction Assay

The setup of the two-choice attraction assay and the method were similar to those of the two-choice assay, except for adding 100 μL of 10% apple cider vinegar (ACV) to the bottom of the microcentrifuge tubes (Figure 1E) [25]. The preference index (PI) was calculated as in the T-maze assay.

### 2.6. Larval Two-Choice Assay

The larval two-choice assay was performed in a Petri dish, as illustrated in Figure 1E. Twenty 3rd-instar larvae were starved on a water-soaked filter paper for 5 h, then transferred to the center of a Petri dish (diameter, 9 cm), which was loaded with an approximately 5 mm deep layer of 2.5% agarose. The Petri dish contained, on one side, a cylinder (diameter, 0.5 cm) of food loaded with oil or odorants and, on the other side, a similar cylinder of food loaded with control hexane (Figure 1G). The number of larvae in each zone was counted after 5 min, and the preference index (PI) was calculated as in the T-maze assay. Each treatment was performed in more than six replicates.

### 2.7. Gas Chromatography–Mass Spectrometry (GC-MS)

Chemical composition of clary sage oil was analyzed by gas chromatography–mass spectrometry (GC-MS) (QP2010 plus, Shimadzu, Japan) and RTX-5MS (30 m × 0.25 mm × 0.25 µm, Shimadzu Corp, Kyoto, Japan). The methodology was adopted from Pedroso et al. [26]. The injector temperature was 250 °C, and the detector temperature was 290 °C. Column temperature was kept at 60 °C for 2 min, increased to 180 °C at a rate of 5 °C/min, and then increased to 280 °C at a rate of 10 °C/min. GC carrier gas was helium, and the flow rate was 1 mL/min. Essential oil was diluted to 1% with ethyl acetate, and the injection volume was 1 μL. Scanning speed was 0.5 scan/s from *m*/*z* 40 to 350, and the split ratio was 1:200. The mass spectrometer was operated using 70 eV ionization energy. With the accurate and sensitive characteristics of GC-MS in determining the major constituents of mixtures, one aliquot was used to analyze the major constituents of clary sage oil.

### 2.8. Electroantennography (EAG)

Three- to five-day-old *D. suzukii* were wedged into the narrow end of a truncated 200 μL pipette tip and mounted on a microscope slide using dental wax. The antenna was kept in a stable and unfolded position by a glass micropipette. The recording glass capillary electrode was capped onto the anterior distal region of one antenna, and the reference recording electrode was inserted into the contralateral eye. Oil and components were diluted in paraffin oil, and 10 µL of solution or control paraffin oil was applied on a 0.2 × 30 mm filter strip, which was inserted into a Pasteur pipette to build a stimulus cartridge. The antenna was exposed to a humid air stream flowing at 20 mL/s, and a stimulus pulse was delivered for 0.5 s at 2 mL/s, which was operated by CS55 (Syntech, Kirchzarten, Germany). EAG signals were generated via IDAC-4 (Syntech, Kirchzarten, Germany), and recordings were analyzed with EAG pro 2000 software (Syntech, Kirchzarten, Germany). More than fifteen flies were tested per compound, and each fly was only tested once. 

### 2.9. Statistical Analysis

All statistical analyses were performed using GraphPad Prism version 9.4.0 (GraphPad software, San Diego, CA, USA). The Shapiro–Wilk test was used to verify the hypothesis of normality. One-way analysis of variance (ANOVA) followed by Tukey’s mean separation test or Kruskal–Wallis test followed by Nemenyi test was used to compare the preference index induced by different dilutions of odorants, depending on whether assumptions for parametric tests were met. Student’s unpaired t-test was used to compare the EAG responses elicited by control and odorants.

## 3. Results

### 3.1. Clary Sage Oil Repels D. suzukii Adults and Larvae

We first developed a T-maze assay to check the effects of clary sage oil on the behavior of *D. suzukii* adults. The results showed that clary sage oil exhibited a significant repellent effect at the 10^−1^ and 10^−2^ dilutions (*v*/*v*), and 10^−3^ (*v*/*v*) essential oil had no repellency (Figure 1A,B). Furthermore, avoidance behavior was also observed in the two-choice assay and two-choice attraction assay; the difference between these two assays was the ACV in the bottom of the microcentrifuge tubes in the latter. Clary sage oil repelled *D. suzukii* at the 0.5 and 10^−1^ dilutions (*v*/*v*) in both assays, while 10^−3^ (*v*/*v*) oil had no effect on the preference of *D. suzukii* (Figure 1C–F). In addition, we conducted a larvae assay to examine the repellency of clary sage oil against *D. suzukii* larvae and found that clary sage oil had a marked deterrent effect against *D. suzukii* larvae for food feeding at the 10^−1^ and 10^−2^ dilutions (*v*/*v*) (Figure 1G,H).

### 3.2. Analysis of the Constituents of Clary Sage Oil 

We analyzed the constituents of clary sage oil through GC-MS. The results showed that clary sage oil consisted of 63 compounds, of which the most abundant constituents were linalyl acetate (40.03%) and linalool (23.02%). The proportions of the other constituents were rather low; most of these constituents were less than 1%, as shown in Table 1. 

### 3.3. Constituents of Clary Sage Oil Repel D. suzukii Adults and Larvae

In order to explore whether the constituents were responsible for the repellent effect of clary sage oil on *D. suzukii*, we conducted assays to separately test the repellent activity of the two major constituents of clary sage essential oil, linalyl acetate and linalool. We found both compounds exhibited repellency in all of the assays, as did clary sage oil. However, the repellent potency was slightly different from that of clary sage oil—the repellent effect of clary sage oil was stronger than that of linalyl acetate and linalool in the T-maze assay (Figure 2A and Figure 3A); contrary to the T-maze assay, linalyl acetate and linalool presented higher repellent activity in both the two-choice assay and the two-choice attraction assay (Figure 2 and Figure 3). Additionally, linalyl acetate and linalool showed comparable repellent effects against *D. suzukii* larvae (Figure 2 and Figure 3).

### 3.4. Clary Sage Oil and Constituents Elicit EAG Responses in D. suzukii Adults

Since clary sage oil and its two major constituents, linalyl acetate and linalool, conferred repellency against *D. suzukii* adults, we wondered whether the repellent effect was mediated by the olfactory system; thus, we examined the EAG responses of the antennae of *D. suzukii* adults against clary sage oil, linalyl acetate, and linalool. Clary sage essential oil elicited a significant EAG response, and the EAG responses were stronger with increased doses of oil (Figure 4). Also, both linalyl acetate and linalool elicited EAG responses, which occurred in a dose-dependent manner; however, the amplitude was much smaller than that of clary sage oil even at the high concentration of 10^−1^ (*v*/*v*) (Figure 4).

## 4. Discussion

Essential oils, derived from various aromatic plants, have emerged as potent and eco-friendly alternatives to traditional chemical pesticides. Compared to synthetic pesticides, one of the key advantages of using essential oils in pest control is their safety for humans and pets. In this study, we found that clary sage oil exhibited a significant deterrent effect against both *D. suzukii* adults and larvae, and two main constituents, linalyl acetate and linalool, were responsible for the deterrent effect via the olfactory system.

Since the tremendous threat to soft-skinned fruit crops caused by *D. suzukii*, spinosad and other broad-spectrum insecticides have been frequently applied for managing *D. suzukii.* Concerning the resistance issue and food safety problems, essential oil, derived from natural plants, is a good candidate solution for *D. suzukii* management [4,7]. In addition to the insecticidal activity, numerous studies reported that essential oils, including lavender oil, catnip oil, mandarin oil, tea tree oil, peppermint oil, ginger oil, geranium oil, white spruce oil, and so on, had excellent deterrent effects against *D. suzukii* [27,28,29]. Moreover, certain essential oils exhibit both toxicity and deterrence against *D. suzukii*, for example, seven species of *Baccharis* [30]. 

In our study, we showed that clary sage oil exhibited a deterrent effect against *D. suzukii*, which was consistent with observations in other pests [21,22,23,24]. Shibuya et al. 2021 reported that clary sage oil elicited attraction behavior in *Drosophila melanogaster* Meigen (Diptera: Drosophilidae) larvae, as did many other essential oils, such as eucalyptus lemon, spikenard, ginger, cypress, pepper, clove, cinnamon cassia, and citronella java [31]. However, our results were discrepant with those published; we found clary sage oil repelled *D. suzukii* larvae. Distinct behaviors induced by natural compounds between *D. melanogaster* and *D. suzukii* are common; as described previously, camphor was a strong repellent against *D. melanogaster*, but it had little repellent activity against *D. suzukii* [32]. Moreover, citronellal was a strong repellent against *D. melanogaster*, *Drosophila pseudoobscura* Frolova (Diptera: Drosophilidae) and *Drosophila virilis* Sturtevant (Diptera: Drosophilidae) but not for *D. suzukii* [33]. 

Our study identified that linalyl acetate and linalool were the two major constituents of clary sage oil, while the other components were less abundant, which was similar to that previously described, where linalyl acetate, linalool, and sclareol were the main components identified in clary sage oil [31]. Recently, Niu reported that clary sage oil consisted of nineteen major chemical constituents, which is considerably less than found in our study, and the most abundant component was isopropyl myristate (28.74%), followed by linalyl acetate (20.07%) and linalool (15.18%) [20]. Although the constituents differ slightly because of the diversity of the oil’s origin and the detection method used, the common point is that linalyl acetate and linalool are the main constituents.

Here, we found two major constituents, linalyl acetate and linalool, exhibited repellent effects against both *D. suzukii* adults and larvae as did clary sage oil. Linalool, one of the monoterpenoids, elicited a deterrence effect against numerous pests, such as *Ae. aegypti* [34,35], *Aedes albopictus* Skuse (Diptera: Culicidae) [34], *Anopheles gambiae* Giles ss (Diptera: Culicidae) [36], *Ixodes ricinus* Linnaeus (Acari: Ixodidae) [37], *Musca domestica* Linnaeus (Diptera: Muscidae) [38], and *Sitophilus oryzae* Linnaeus (Coleoptera: Curculionidae) [39]. However, Dewitte et al. (2021) found that the constituents of blackberry (*Rubus fruticosus*), acetaldehyde, hexyl acetate, linalool, myrtenol, L-limonene, and camphene, were significantly attractive to *D. suzukii* [40]. Another paper reported that (R)-(+)-limonene possessed strong repellency against *D. suzukii*; moreover, other terpenoids, menthone, (+)-fenchone menthyl acetate, nerol, (−)-α-thyjone, and norcamphor, all elicited repellency in *D. suzukii* [32]. The discrepancies between these studies, including our study, might be caused by the concentration used in the behavioral assays; since insects have sensitive olfactory systems, small differences including the methods and flies in the assays might lead to distinct behaviors. Compared with linalool, few studies have been conducted on the activity of linalyl acetate. A previous study indicated that linalyl acetate was not only the most abundant constituent of essential oils but also conferred high repellent activity against *Tetranychus urticae* Boisduval (Acari: Tetranychidae) [22]. In our study, *D. suzukii* showed avoidance of linalool and linalyl acetate in all of the behavioral assays we conducted; as the major constituents of clary sage oil, we suggest that these two compounds might be responsible for the repellent activity of clary sage oil against *D. suzukii*.

Insects rely heavily on their sense of smell for various behaviors such as finding food and mates, as well as avoiding predators. The antennae of insects are equipped with numerous olfactory receptor neurons that are specialized for detecting volatile chemical compounds in the air [41,42]. EAG responses contribute to the understanding of how different odorants are encoded in the insect’s nervous system. Earlier, linalool has been shown to induce marked EAG responses in *D. suzukii* [43]. Here, we observed that clary sage oil and its two constituents all elicited an EAG response in *D. suzukii*, which means these compounds were perceived by the fly antenna via activation of the olfactory system to elicit avoidance behavior. Additionally, the EAG response to clary sage oil was bigger than that to linalool and linalyl acetate, indicating that the olfactory system was sensitive to clary sage oil. The behavior assay showed both linalool and linalyl acetate exhibited comparable repellent effects to clary sage oil, which might be because behavior is complicated and involves both the detection of sensory input by the peripheral nervous system and then signal processing by the central nervous system.

## 5. Conclusions

In summary, clary sage oil and its two major constituents, linalool and linalyl acetate, evoke an olfactory response and elicit repellency in *D. suzukii* adults and larvae. Our study not only provides evidence that clary sage oil and its major constituents could be possible alternatives for *D. suzukii* control but also lays a foundation for further mechanistic studies of essential oil repellency.

## Figures and Tables

**Figure 1 insects-15-00733-f001:**
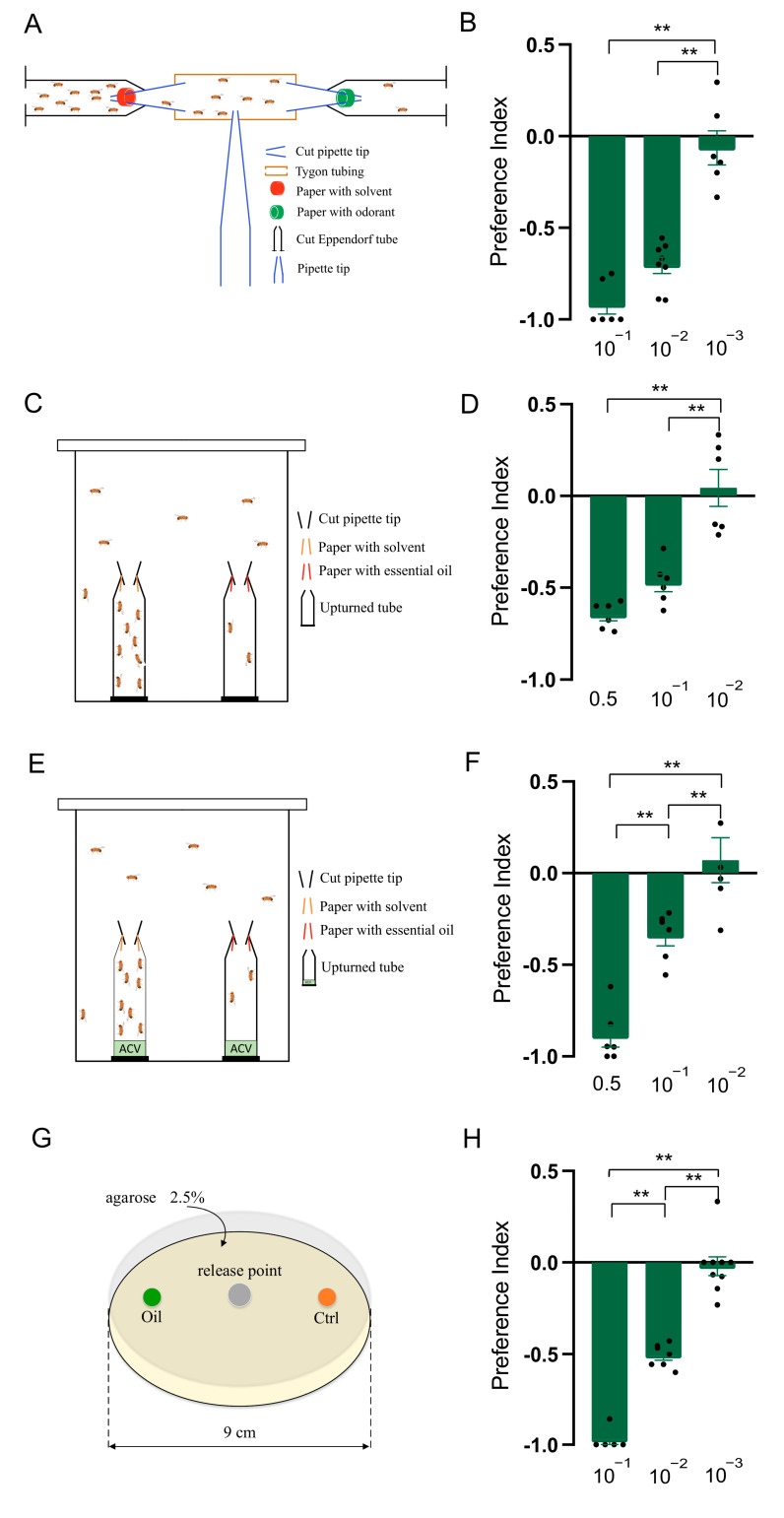
Clary sage oil elicited repellency in *Drosophila suzukii*. (**A**) Schematic drawing of T-maze assay. (**B**) Preference index of *D. suzukii* for clary sage oil in T-maze assay. (**C**) Schematic drawing of two-choice assay. (**D**) Preference index of *D. suzukii* for clary sage oil in two-choice assay. (**E**) Schematic drawing of two-choice attraction assay. (**F**) Preference index of *D. suzukii* for clary sage oil in two-choice attraction assay. (**G**). Schematic drawing of larval two-choice assay. (**H**) Preference index of *D. suzukii* larvae for clary sage oil. Data are plotted as mean ± s.e.m. Dots denote the value of each repeat. Statistical analysis was conducted by using one-way analysis of variance (ANOVA) followed by Tukey’s means separation test in (**D**,**F**), and Kruskal–Wallis followed by Nemenyi test in (**B**,**H**). Asterisks indicate a significant difference between the preference index of compounds at different dilutions at ** *p* < 0.01.

**Figure 2 insects-15-00733-f002:**
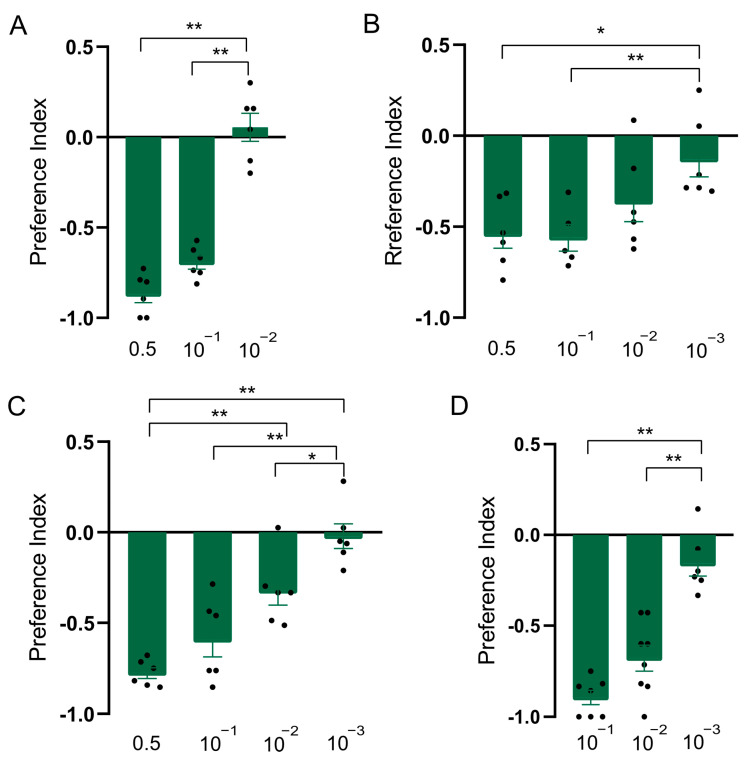
Repellency elicited by linalyl acetate in *Drosophila suzukii*. (**A**) Preference index of *D. suzukii* for linalyl acetate in T-maze assay. (**B**) Preference index of *D. suzukii* for linalyl acetate in two-choice assay. (**C**) Preference index of *D. suzukii* for linalyl acetate in two-choice attraction assay. (**D**) Preference index of *D. suzukii* larvae for linalyl acetate. Data are plotted as mean ± s.e.m. Dots denote the value of each repeat. Statistical analysis was conducted by using one-way analysis of variance (ANOVA) followed by Tukey’s means separation test. Asterisks indicate a significant difference between preference indices of compounds at different dilutions at * *p* < 0.05 and ** *p* < 0.01.

**Figure 3 insects-15-00733-f003:**
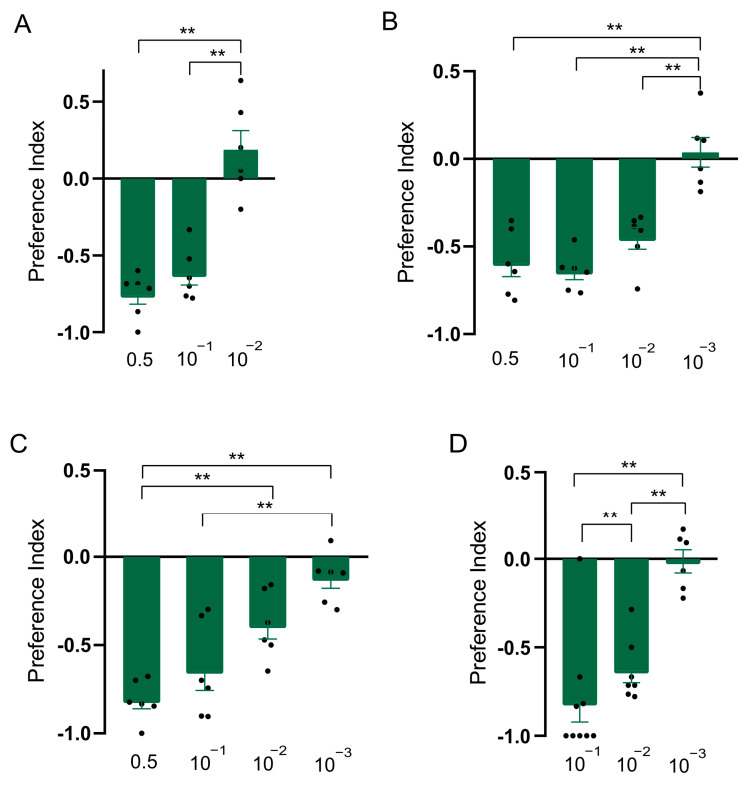
Repellency elicited by linalool in *Drosophila suzukii*. (**A**) Preference index of *D. suzukii* for linalool in T-maze assay. (**B**) Preference index of *D. suzukii* for linalool in two-choice assay. (**C**) Preference index of *D. suzukii* for linalool in two-choice attraction assay. (**D**) Preference index of *D. suzukii* larvae for linalool. Data are plotted as mean ± s.e.m. Dots denote the value of each repeat. Statistical analysis was conducted by using one-way analysis of variance (ANOVA) followed by Tukey’s means separation test in (**A**–**C**) and Kruskal–Wallis followed by Nemenyi test in (**D**). Asterisks indicate a significant difference between preference indices of compounds at different dilutions at ** *p* < 0.01.

**Figure 4 insects-15-00733-f004:**
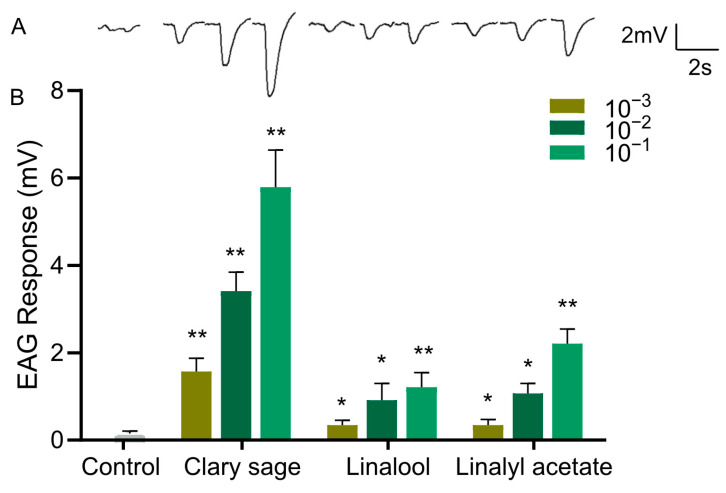
EAG responses of *Drosophila suzukii* adults to clary sage oil and its two major constituents. (**A**) Representative EAG traces elicited by clary sage oil, linalool, and linalyl acetate at 10^−1^ dilution. (**B**) EAG responses to clary sage oil, linalool, and linalyl acetate. *n* ≥ 15 flies. Data are plotted as mean ± s.e.m. Statistical analysis was conducted by Student’s unpaired t-test, and asterisks indicate a significant difference between the response for control and odorants at * *p* < 0.05 and ** *p* < 0.01.

**Table 1 insects-15-00733-t001:** Top twenty abundant chemical constituents in clary sage oil. Relative proportions of the essential oil constituents are expressed as percentages.

No.	Retention Time (Min)	Compounds	CAS	Relative Percentage (%)
1	21.77	Linalyl acetate	115-95-7	40.03
2	21.04	Linalool	78-70-6	23.02
3	28.56	Geranyl acetate	105-87-3	3.35
4	27.75	Neryl acetate	141-12-8	2.2
5	19.96	α-Cubenene	17699-14-8	1.8
6	30.2	Linalyl formate	115-99-1	1.11
7	31.2	Isotridecan-1-ol	27458-92-0	1.03
8	16.69	Cis-linalool oxide	14009-71-3	1.02
9	7.4	β-Pinene	127-91-3	0.96
10	17.78	2-Furanmethanol	34995-77-2	0.93
11	23.39	β-Caryophyllene	87-44-5	0.92
12	19.1	L-Camphor	464-48-2	0.9
13	29.94	Dihydro-beta-ionone	17283-81-7	0.87
14	10.17	Ocimene	3338-55-4	0.73
15	28.77	(−)-Isoledene	95910-36-4	0.71
16	26.33	α-Terpineol	98-55-5	0.68
17	29.25	Nerol	106-25-2	0.64
18	22.76	(−)-Terpinen-4-ol	20126-76-5	0.59
19	10.46	1,5,8-p-Menthatriene	21195-59-5	0.53
20	26.61	γ-Terpineol	586-81-2	0.48

## Data Availability

The data presented in this study are available upon request from the corresponding author.

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
