# Peer review of "Deterrent Effects of Clary Sage Oil and Two Major Constituents against Drosophila suzukii (Diptera: Drosophilidae)"

_insects, 2024, doi:10.3390/insects15100733_

Round 1

Reviewer 1 Report

Comments and Suggestions for Authors

To identify natural "insecticides" for Drosophila suzukii, an invasive species that affects soft-skinned fruits, this manuscript titled “Deterrent Effects of Clary Sage Oil and Two Major Constituents Against Drosophila suzukii” investigates whether sage oil and its major chemical components could potentially serve as odor repellents. Using T-maze and two-choice assays, the authors demonstrated that D. suzukii exhibited repellent behavior towards clary sage oil in a dosage-dependent manner. Through GC-MS analysis, they identified linalyl acetate and linalool as the two major constituents of clary sage oil and further confirmed that D. suzukii displayed similar repellent behavior to these compounds, albeit at higher concentrations. Finally, the authors showed that these chemical components triggered electrophysiological spikes in D. suzukii antennae, suggesting that the observed behavior was mediated by the olfactory system.

The discovery of a natural repellent is crucial for pest control in agriculture. While there are some interesting findings in this paper, several significant improvements are necessary before it can be published:

Major Concerns: 

1. Quantification of Repellency Index: The quantification of the repellency index is confusing. Based on the formula provided in section 2.3, Figures 1-3 show positive repellency indexes in adults but negative ones in larvae, suggesting that the odor (sage oil) might actually attract D. suzukii in adult flies. If this is true, sage oil may not serve as an effective repellent for adult flies.

 2.Effectiveness of Major Components: Although linalyl acetate and linalool are major components of sage oil, neither of them could trigger behavior similar to the original sage oil at the same concentration. One hypothesis could be that these two components activate distinct olfactory receptors, and both function conjunctively or in parallel to contribute to the repellent behavior. This could be tested by applying a mixture of linalyl acetate and linalool. If this mixture fails to mimic the behavior observed with sage oil, it suggests that other components, despite their smaller percentage in the oil, play a key role in repelling the flies.

 3.Lack of Controls: For EAG (Electroantennography) recordings, a negative control with air should be applied to validate that the spikes observed in Figure 4 are not due to contamination. 

4.Statistical Analysis: In Figures 1-4, statistical analysis is required to assess whether the observed differences are statistically significant.

Author Response

1. Quantification of Repellency Index: The quantification of the repellency index is confusing. Based on the formula provided in section 2.3, Figures 1-3 show positive repellency indexes in adults but negative ones in larvae, suggesting that the odor (sage oil) might actually attract D. suzukii in adult flies. If this is true, sage oil may not serve as an effective repellent for adult flies.

Response: Thank you for pointing this out, it’s a mistake, now we revised it in the method section and Figures 1-3. We defined it using preference index (PI) in all behavioral assays. “The Preference Index (PI) was calculated as (O-C)/(O+C), where O is the number of flies in the test odorant trap, C is the number of flies in the solvent control trap.” The results showed clary sage oil and two constitutes all exhibited repellent effects against D. suzukii adults and larvae as shown in Figures 1-3.

2. Effectiveness of Major Components: Although linalyl acetate and linalool are major components of sage oil, neither of them could trigger behavior similar to the original sage oil at the same concentration. One hypothesis could be that these two components activate distinct olfactory receptors, and both function conjunctively or in parallel to contribute to the repellent behavior. This could be tested by applying a mixture of linalyl acetate and linalool. If this mixture fails to mimic the behavior observed with sage oil, it suggests that other components, despite their smaller percentage in the oil, play a key role in repelling the flies.

Response: Thank you for your suggestion, we totally agree with your point, sometimes the constitute with small percentage would play a key role, for example, in the insect sex pheromone. Other constitutes also conferred the avoidance of suzukiipublished in previous papers, for example nerol and camphor, even though the percentage of these compounds was low in our clary sage oil, they would confer certain function in clary sage oil repellency. Here, we found the repellent potency of linalyl acetate and linalool were weaker than clary sage oil in T-maze, however, the repellent effects were stronger induced by linalyl acetate and linalool in the Two-choice assay and Two-choice attraction assay. The results showed that the mixture and the pure slightly had distinct repellent potency, which is common in the biological activity of mixtures. This manuscript did not claim that other constitutes were not involved in the repellency, and since we don’t have enough flies in a short time, so we haven’t add this experiment.

3. Lack of Controls: For EAG (Electroantennography) recordings, a negative control with air should be applied to validate that the spikes observed in Figure 4 are not due to contamination.

Response: As suggested, we modified the Figure 4, now the EAG response values and EAG trace of control were showed in Figure 4, the solvent control elicited tiny EAG response.

4.Statistical Analysis: In Figures 1-4, statistical analysis is required to assess whether the observed differences are statistically significant.

Response: As suggested, we added the statistical analysis in Figures 1-4.

Reviewer 2 Report

Comments and Suggestions for Authors

In my opinion, this research on the deterrent effect of clary sage oil and two major compounds against D. suzukii is interesting. I think the research seems to be well conducted and the conclusions justified by the results. However, the manuscript lacks statistical details in the Material and Methods and also in the figures. This should be included before publication. Additional aspects such as expanding some parts of the Introduction and Discussion could improve the quality of this manuscript. Also, writing in a more quantitative way, highlighting the most significant differences. Formatting and English should be checked. Details are given in the attached file.

Comments on the Quality of English Language

I think only minor English edition is required.

Author Response

  1. Compete the name of EAG

Response: We added the full name of EAG.

  1. Abstract is mainly focused in objectives. In my opinion, you should include the most significant quantitative data.

Response: Thanks for your suggestion, we modified the abstract.

  1. Please, include unitsof percentage in the abstract.

Response: We added the units “%”.

  1. I think the structure of the introduction is good, but it's a bit short. You could include more examples and develop your exposition a bit more.

Response: Thanks for your suggestion,  since the some content of discussion and introduction was overlap, we modified the introduction.

  1. Please, explain the most relevant advancements in the use of natural products against this pest.

Response: Now there is no commercial product, because these research is still in the lab stage, so we described that essential oil is a good candidate solution for D. suzukii management.

  1. Examples should be better if they are from other Brachycera.

Response: We modified this paragraph, and added more examples.

  1. You should include an additional point on Statistics.

Response: We added the statistical analysis in the figure legends.

  1. Long hyphenwith the dilutions

Response: Thanks, we revised it.

  1. Figure (Please, check formatting)

Response: We check the formatting of the whole manuscript and revised them.

  1. It would be good if you could enlarge the images and use more distinctive colours. You should include statistical information, including also number of repetitions for essay.

Response: The Figure 1 was enlarged, and also the statistical analysis was added. The figure legend shows dots denote the value of each repeat, so one dot means one repeat.

  1. Please, include the statistical information.

Response: As suggested, we added the statistical analysis.

  1. Please, include the statistical information, also significant differences in the graphs.

Response: As suggested, we added the statistical analysis.

  1. Please, include the statistical information.

Response: As suggested, we added the statistical analysis.

Reviewer 3 Report

Comments and Suggestions for Authors

The manuscript delves into crucial information about the repellent action of a species of Lamiaceae and its major compounds, underscoring the significance of the author's work in this field.

The authors should change the keywords. By using different keywords from the title, you can attract a broader range of readers who might be interested in your research.

It's crucial to keep the readers updated on the spread of Drosophila suzukii to Latin America, Africa, and Oceania. Authors should ensure this information is accurately reflected in their abstracts and introductions.

When citing for the first time, authors should include authority, order, and family.

The origin of the first insects of the D. suzukii colony should be more detailed. Clear information about the local host (city, country, latitude, and longitude) and the year is essential to maintain the manuscript's credibility.

Figures 1a, 1E, and 1G should be presented in the methodology separately from the results to provide a clear understanding of the experimental setup before the results are discussed.

It's essential to provide clear and precise information about the number of replicates and the distribution of males and females in each experiment. The authors' role in providing accurate data is crucial for the scientific community.

The discussion should focus on flies of the genus Drosophila, such as Drosophila melanogaster and Drosophila suzukii, and plants of the Lamiaceae family. Essential oils of Myrtaceae and Lamiaceae are the most widely used in the control of D. suzukii, so discussing plants that are so phylogenetically distant is unnecessary.

I've included more suggestions in the attached text.

Author Response

The manuscript delves into crucial information about the repellent action of a species of Lamiaceae and its major compounds, underscoring the significance of the author's work in this field.

  1. The authors should change the keywords. By using different keywords from the title, you can attract a broader range of readers who might be interested in your research.

Response: As suggested, we revised the keywords into “Spotted-Wing Drosophila; Clary sage oil; Linalyl acetate; Linalool; Repellency; EAG”.

  1. It's crucial to keep the readers updated on the spread of Drosophila suzukii to Latin America, Africa, and Oceania. Authors should ensure this information is accurately reflected in their abstracts and introductions.

Response: Thanks, we checked the manuscript and revised it.

  1. When citing for the first time, authors should include authority, order, and family.

Response: As suggested, we checked the whole manuscript and added the detail information.

  1. The origin of the first insects of the D. suzukii colony should be more detailed. Clear information about the local host (city, country, latitude, and longitude) and the year is essential to maintain the manuscript's credibility.

Response: we added the detail information of the origin of D. suzukii in the method section. “D. suzukii were collected from field in Haikou, Hainan Province, China (20 °N, 110°E) in 2018, which were kindly provided by Shaoying Wu (Hainan Univ.). ”

  1. Figures 1a, 1E, and 1G should be presented in the methodology separately from the results to provide a clear understanding of the experimental setup before the results are discussed.

Response: Thanks for your suggestion, Figure 1A, 1C, 1E and 1G were the schematic drawing of different behavioral assay, they were belong to the methodology, but we think it might be appropriate to put the schematic drawings here, which is convenient for readers to understand the results corresponding to the certain behavioral assays.

  1. It's essential to provide clear and precise information about the number of replicates and the distribution of males and females in each experiment. The authors' role in providing accurate data is crucial for the scientific community.

Response: As suggested we added the number of males and females in the Two-choice assay and Two-choice attraction assay. “Forty to fifty three- to five-day-old D. suzukii (half males and half females) were gently introduced into the glass beaker and covered with gauze ”. For replicates in each bioassay, the figure legend shows dots denote the value of each repeat, so one dot means one repeat, and in EAG recording, n≥15 flies.

  1. The discussion should focus on flies of the genus Drosophila, such as Drosophila melanogaster and Drosophila suzukii, and plants of the Lamiaceae family. Essential oils of Myrtaceae and Lamiaceae are the most widely used in the control of D. suzukii, so discussing plants that are so phylogenetically distant is unnecessary.

Response: Thanks for your suggestion, since the effect of essential oil was already in the introduction, so we delete this paragraph, and modified it.

  1. I've included more suggestions in the attached text.

Response: Thanks for you suggestion, we revised as suggested as listed one by one.

Reviewer 4 Report

Comments and Suggestions for Authors

 Wang et al. report the results of laboratory assays on the effect of an essential oil on Drosophila suzukii repellence/attraction.

MAJOR POINTS

         I.            There is considerable information missing from the methods sections; it would not be possible to repeat this study with the information provided by the authors.

       II.            The authors need to justify why some of the assays were performed over such long periods (24 h) given that insects normally respond to olfactory stimuli within seconds or minutes.

     III.            Why were no statistical analyses performed?

I have written suggestions and numbered points on a scanned copy of the manuscript.

NUMBERED POINTS (see scanned file)

1. I could not understand this text, please reword for clarity.

2. Do not use keywords that are already in the title.

3. You only have limited laboratory evidence for potential as a control agent; please tone down your assertion that the oil may have potential in pest management.

4a. Please given details of the poplar wool and filter paper content of the diet.

4b. What solvent was used to dilute the oil? Was solvent allowed to evaporate, for how long?

5. Were insects tested individually or in groups (section 2.3, 2.4, 2.5)?

6. Did all insects respond? Or how did you classify non-responsive insects?

7. 24 h is a VERY long period over which to evaluate insect responses!  Why so long? In my experience most such assays last a few minutes.

What about olfactory habituation during such a long exposure period?

8. How was the apparatus illuminated?

9. Did you switch the sides of the oil and solvent tubes after each test?

10. Was the apparatus cleaned after each test? How?

11. Section 2.5: Why was the assay performed again with ACV? What was the purpose of this test?

12. If the Petri dish was FILLED with agarose, was there enough room for insects? Do you mean that the dish had a LAYER of agarose? If so, how deep?

13. Was the solvent given time to evaporate prior to testing?

14. Counted for 5 mins? Do you mean "counted after 5 mins"?

15. section 2.6 – was this assay replicated? Details missing.

16.  adapted or adopted? Please clarify.

17. Was the GC-MS assay replicated?

18. 15 repeats? Do you mean that 15 insects were tested? Please clarify. Was each insect tested only once (a single pulse?) or more times?

19. Please increase the size of Figure 1 to improve readability.

20. Clarify that each component was tested separately (not in mixtures).

21. The A and B letters are missing from Figure 4.

22. The influence of essential oils on insects was already mentioned in the Introduction. Delete this paragraph.

23. If this is true, how many essential oil based products are currently registered against D. suzukii in China?

24. This text should mark the beginning of the Discussion because it is the start of the text where you mention your results.

25. Shibuya reference should be numbered.

26. The authors should provide their raw data as supplementary material (Excel file) following MDPI guidelines.

27. The references should be formatted following Insects guidelines.

Comments on the Quality of English Language

Many grammatical errors. Needs editing for clarity.

Author Response

Wang et al. report the results of laboratory assays on the effect of an essential oil on Drosophila suzukii repellence/attraction.

MAJOR POINTS

  1. There is considerable information missing from the methods sections; it would not be possible to repeat this study with the information provided by the authors.

Response: As your suggestion, we added the detail information in the methods sections.

  1. The authors need to justify why some of the assays were performed over such long periods (24 h) given that insects normally respond to olfactory stimuli within seconds or minutes.

Response: Some insects respond to olfactory stimuli, but from our observation flies need more time to make the choice, In the olfactory apparatus, flies were habituated then made the choice, and after they made the choice to enter two arms, they would not be allowed to return based on the structure of olfactory apparatus. So we referred previous papers and counted the number of flies in the two arms after 24h.

What about olfactory habituation during such a long exposure period?

Response: In the olfactory apparatus, flies were habituated then made the choice, and after they made the choice to enter two arms, they would not allow to return based on the structure of olfactory apparatus.

  1. Why were no statistical analyses performed?

Response: We added the statistical analyses in Figure 1-4.

I have written suggestions and numbered points on a scanned copy of the manuscript.

NUMBERED POINTS (see scanned file)

  1. I could not understand this text, please reword for clarity.

Response: This sentence was deleted, since abstract was too long, so we only presented the main results.

  1. Do not use keywords that are already in the title.

Response: As suggested, we revised the keywords into “Spotted-Wing Drosophila; Clary sage oil; Linalyl acetate; Linalool; Repellency; EAG”.

3.You only have limited laboratory evidence for potential as a control agent; please tone down your assertion that the oil may have potential in pest management.

Response: As suggested, we modified this sentence, “This study illustrated that clary sage oil and the major constitutes might have a great potential for management of D. suzukii.”

4a. Please given details of the poplar wool and filter paper content of the diet.

Response: As your suggestion, we added the detail information of poplar wood wool and filter paper, “D. suzukii were reared on standard cornmeal–yeast agar medium (Genesee Scientific, San Diego, CA, USA) with poplar wood wool (Hualeji, Hangzhou, Zhejiang, China) and filter paper (Jiaojie, Dongguan, guangdong, China). ”

4b. What solvent was used to dilute the oil? Was solvent allowed to evaporate, for how long?

Response: The solvent control used in T-maze assay was paraffin oil, we added in the text. The solvent was not allowed to evaporate, since the filter paper was lined the wall of the microcentrifuge tube and paraffin oil was sticky, so that 50 μL odorant or solvent were not able to affect the behavior of flies, also the odorant would evaporate slowly.

5.Were insects tested individually or in groups (section 2.3, 2.4, 2.5)?

Response: Twenty flies, including 10 males and 10 females, were tested as a replicate, we modified in the text, “Twenty three- to five-day-old D. suzukii flies (10 males and 10 females) were gently introduced into the Tygon tubing through the small hole made in the middle of the tubing. ” 

  1. Did all insects respond? Or how did you classify non-responsive insects?

Response: Not all insects were responded to odorants, flies did not enter each arm and stayed in the Tygon tubing that were counted as non-responsive flies. We calculated Preference Index (PI) using the number of flies entering each arm, not the  non-responsive flies.

  1. 24 h is a VERY long period over which to evaluate insect responses!  Why so long? In my experience most such assays last a few minutes.

Response: Yes, 24 h is a long period, Some insects respond to olfactory stimuli, but from our observation flies need more time to make the choice, In the olfactory apparatus, flies were habituated then made the choice, and after they made the choice to enter two arms, they would not be allowed to return based on the structure of olfactory apparatus. So we referred previous papers and counted the number of flies in the two arms after 24h.

What about olfactory habituation during such a long exposure period?

Response: In the olfactory apparatus, flies were habituated then made the choice, and after they made the choice to enter two arms, they would not allow to return based on the structure of olfactory apparatus.

  1. How was the apparatus illuminated?

Response: The olfactory apparatus was put in the growth chamber, so the illumination is the same as the insect rearing.

  1. Did you switch the sides of the oil and solvent tubes after each test?

Response: No, we did not switch the sides of the oil and solvent tubes, but many replicates were conducted at one time, and the orientations of different replicates were not the same, so it was equal to switch the sides.   

  1. Was the apparatus cleaned after each test? How?

Response: The apparatus was disposable, we made them prior the experiments and one apparatus only used for one replicate.

  1. Section 2.5: Why was the assay performed again with ACV? What was the purpose of this test?

Response: The difference between two-choice assay and two-choice attraction assay was the absence and presence of ACV, flies were attracted by ACV, so this assay was conducted to determine the response of flies in the presence of food ACV.

  1. If the Petri dish was FILLED with agarose, was there enough room for insects? Do you mean that the dish had a LAYER of agarose? If so, how deep?

Response: Yes, the dish had a layer of agarose, the Petri dish (diameter, 9 cm) filled with 2.5% agarose, and after the agarose coagulation, the larvae placed were on the agarose. The deep of the agarose was about 0.5 cm.

  1. Was the solvent given time to evaporate prior to testing?

Response: The solvent was not allowed to evaporate, since the filter paper was lined the wall of the microcentrifuge tube and paraffin oil was sticky, so that 50 μL odorant or solvent were not able to affect the behavior of flies, also the odorant would evaporate slowly.

  1. Counted for 5 mins? Do you mean "counted after 5 mins"?

Response: Count after 5 mins. We revised in the text.

  1. section 2.6 – was this assay replicated? Details missing.

Response: The figure legends show dots denote the value of each repeat, one dot means one repeat, so we did not described the repeat in the method, it’s in the figure legends.

  1. adapted or adopted? Please clarify.

Response: It’s adopted. The method was modified from Pesroso et al. We revised this sentence, “The methodology was adopted from Pedroso et al”.

  1. Was the GC-MS assay replicated?

Response: The identification of clary sage oil by GC-MS was one replication, since GC-MS is a very accurate and sensitive setup, and the purpose is to determine the major constitutes of oil, in general, one replication is enough to identify the constitute and percentage of a mixture.

  1. 15 repeats? Do you mean that 15 insects were tested? Please clarify. Was each insect tested only once (a single pulse?) or more times?

Response: Yes, each insect only tested once, and more than 15 insects were tested individually in EAG experiment.

  1. Please increase the size of Figure 1 to improve readability.

Response: As your suggestion, the Figure 1 was enlarged.

  1. Clarify that each component was tested separately (not in mixtures).

Response: As suggested, we modified it, “we conducted the repellent activity of two major constitutes of clary sage essential oil, linalyl acetate and linalool, respectively.” 

  1. The A and B letters are missing from Figure 4.

Response: Thanks for pointing this out, we added A and B in the Figure 4.

  1. The influence of essential oils on insects was already mentioned in the Introduction. Delete this paragraph.

Response: As your suggestion, we modified this paragraph.

  1. If this is true, how many essential oil based products are currently registered against D. suzukii in China?

Response: Now there is no commercial product, because these research is still in the lab stage, so we described that essential oil is a good candidate solution for D. suzukii management.

  1. This text should mark the beginning of the Discussion because it is the start of the text where you mention your results.

Response: Thanks for your suggestion, we modified the discussion section, we described the results at the beginning of discussion, and this part is the comparasion of our results with others.

  1. Shibuya reference should be numbered.

Response: As suggested, this reference was numbered.

  1. The authors should provide their raw data as supplementary material (Excel file) following MDPI guidelines.

Response: As your suggestion, we added it “Data Availability Statement: The data presented in this study are available upon request from corresponding author.”

  1. The references should be formatted following Insects guidelines.

Response: We revised the format of references.

Round 2

Reviewer 1 Report

Comments and Suggestions for Authors

I'm happy to see that most of my suggestions were taken by the authors and have no further questions prior to its publication.  

Author Response

No comments were found.

Reviewer 3 Report

Comments and Suggestions for Authors

Many of the suggestions and corrections were not made by the authors, such as:

The inclusion of the last name of the classifying author (authority) when citing a species for the first time,

The detailing of how many generations or years the D. suzukii colony reared in laboratory conditions,

The names of the genera should not be abbreviated in the captions,

Figures 1A, 1C, 1E, and ! g should be presented in the materials and methods section and not in the results section

Considering this work's potential impact, I would like to encourage the authors to review this information carefully. Your research has the potential to significantly contribute to our understanding of D. suzukii and related topics, inspiring further exploration in this field.

Author Response

  1. The inclusion of the last name of the classifying author (authority) when citing a species for the first time.

Response: As suggested, we added the last name of the classifying author. 

  1. The detailing of how many generations or years the D. suzukii colony reared in laboratory conditions.

Response: We modified the origin of D. suzukii colony, “D. Suzukii, kindly provided by Shaoying Wu (Hainan Univ.), were collected from field in Haikou, Hainan Province, China (20 °N, 110°E) in 2018, which were reared in laboratory conditions for more than five years. ”

  1. The names of the genera should not be abbreviated in the captions

Response: As your suggestion, we revised the caption of figures.

  1. Figures 1A, 1C, 1E, and ! g should be presented in the materials and methods section and not in the results section.

Response: Thanks for your suggestion, we totally agree that the Figures 1A, 1C, 1E and 1G are the methods, it could be in the materials and methods section, but we thought that the schematic drawing of behavioral assay besides the results will be helpful to readers to compare the differences of methods in four behavioral assays, and compare the repellent effects elicited by odorants in different behavioral assays. In addition, we added Figures 1A, 1C, 1E and 1G in the method section after description of method details. But if you insist, we would like to move them to materials and methods section as a separate figure.

Reviewer 4 Report

Comments and Suggestions for Authors

The authors have improved their manuscript but important issues remain that the authors need to address.

1. How were negative values compared statistically with positive values in Fig 1, Fig 2 and Fig 3? How was this possible?

2. Did the data meet the assumptions of ANOVA? I spotted possible issues of inequality of variances in the figures. The authors need to state how they tested for homoscedasticity.

3. Section 2.7. The authors need to make it clear that the GC-MS assay was not replicated.

4. Fig 4. Why were treatments compared by t-test? This seems inappropriate because the experimental design is that of ANOVA or GLM.

5. Illumination of the olfactory test apparatus is not mentioned. This is important because light affects insect responses.

6. Section 2.6. The fact that the agarose layer was 5 mm deep should be mentioned (i.e. the dish was not "filled" with agarose).

7. Section 2.6 should state the number of replicates performed.

8. The Materials and methods section should include a section on the statistical analyses that the authors have now performed.

9. L169 should explain that 15 insects were tested; each insect was tested once and discarded.

10. The authors should format their references following MDPI guidelines.

Comments on the Quality of English Language

Minor editing required.

Author Response

  1. How were negative values compared statistically with positive values in Fig 1, Fig 2 and Fig 3? How was this possible?

Response: The positive value means that flies prefer odorant trap, and the negative value means that flies prefer control trap. High concentrations of odorants repell flies to the control trap, but when the odorant concentration is low, the flies randomly enter the odorant or control trap, so both positive and negative values exist. Since the positive and negative values represent the preference of flies, so when statistical analysis was conducted, all data were analyzed.

  1. Did the data meet the assumptions of ANOVA? I spotted possible issues of inequality of variances in the figures. The authors need to state how they tested for homoscedasticity.

Response: Thank you for pointing this out, we analyzed it again, and found that Figure 1 B and H, Figure 3D, did not meet the assumptions of ANOVA, so we changed the statistical analysis, the detail was described in the method sections 2.9 and in the figure legends.

  1. Section 2.7. The authors need to make it clear that the GC-MS assay was not replicated.

Response: we added one sentence in section 2.7 to clarify it, “With the accurate and sensitive characteristics of GC-MS in determining the major constitutes of mixtures, one aliquot was used to analyze the major constitutes of clary sage oil.”

  1. Fig 4. Why were treatments compared by t-test? This seems inappropriate because the experimental design is that of ANOVA or GLM.

Response: We compared the EAG responses elicited by control and odorants, not the EAG responses elicited by different dilutions of odorants, so we used the student's unpaired t-test to determine two sets of data.

  1. Illumination of the olfactory test apparatus is not mentioned. This is important because light affects insect responses.

Response: We added the illumination of the behavioral assays “The assay was run at 25ËšC with a light/dark photoperiod of 12 h each. ”

  1. Section 2.6. The fact that the agarose layer was 5 mm deep should be mentioned (i.e. the dish was not "filled" with agarose).

Response: As suggested, we added this information in the method, “Twenty 3rd instar larvae were starved on a water-soaked filter paper for 5 h, then transferred to the center of a Petri dish (diameter, 9 cm) which was loaded with approximately 5 mm deep layer of 2.5% agarose.”

  1. Section 2.6 should state the number of replicates performed.

Response: As suggested, we added the number of replicates in the section 2.6, Each treatment was performed more than six replicates.

  1. The Materials and methods section should include a section on the statistical analyses that the authors have now performed.

Response: As your suggestion, we added the statistical analysis section in the method, “All statistical analysis were performed using GraphPad Prism version 9.4.0 (GraphPad software, San Diego, CA, USA). Shapiro—Wilk test was used to verify the hypothesis of normality. One-way analysis of variance (ANOVA) followed by Tukey's means separation test or Kruskal-Wallis followed by Nemenyi test was used to compare the preference index induced by different dilutions of odorants, depending on whether assumptions for parametric tests were met. Student's unpaired t-test was used to compare the EAG responses elicited by control and odorants. ”

  1. L169 should explain that 15 insects were tested; each insect was tested once and discarded.

Response: We modified it in section 2.8, “More than fifteen flies were conducted per compound, and each fly was only tested once. ”

  1. The authors should format their references following MDPI guidelines.

Response: We checked the format of references.

Round 3

Reviewer 3 Report

Comments and Suggestions for Authors

Line 50 authors must include the citation for the information D. suzukii, originated in East Asia, has spread to other parts of the world, including Europe, Americas, Africa and Oceania,

Author Response

Line 50 authors must include the citation for the information D. suzukii, originated in East Asia, has spread to other parts of the world, including Europe, Americas, Africa and Oceania.

Response: As suggested, we added this reference.

Reviewer 4 Report

Comments and Suggestions for Authors

The authors have addressed my concerns.

Comments on the Quality of English Language

minor editing

Author Response

minor editing on language.

Response: We checked the whole manuscript, and revised the editing.